# Forces and Specific Energy of Polyamide Grinding

**DOI:** 10.3390/ma14175041

**Published:** 2021-09-03

**Authors:** Roberto Spina, Bruno Cavalcante, Marco Massari, Roberto Rutigliano

**Affiliations:** 1Dipartimento di Meccanica, Matematica e Management, Politecnico di Bari, 70125 Bari, Italy; roberto.spina@poliba.it; 2Istituto Nazionale di Fisica Nucleare (INFN)—Sezione di Bari, 70125 Bari, Italy; 3Consiglio Nazionale delle Ricerche—Istituto di Fotonica e Nanotecnologie (CNR-IFN), 70126 Bari, Italy; 4Bosch Tecnologie Diesel S.p.A., 70026 Modugno, Italy; marco.massari2@it.bosch.com (M.M.); roberto.rutigliano@it.bosch.com (R.R.)

**Keywords:** grinding, precision mechanics, machining, polyamide, GFRP, material characterization

## Abstract

This work investigated the grinding process of reinforced and nonreinforced polyamide materials using an Al_2_O_3_ grinding wheel. Samples were ground using a custom-made setup of sensors to evaluate in-line temperature, forces, and power. The surface roughness and images were acquired to assess the quality of the final products. The novelty of the work is to correlate the energy evaluation with the process efficiency during processing. Grinding at high cutting depths achieves good surface quality indicators, such as *R_a_* < 5 μm and *R_z_* < 5 μm. Results also reveal that special attention should be given to the infeed speed when cutting unfilled materials to produce good results. With high values of energy partition, the specific grinding energy stabilizes around 60 J/mm³. Strains must be applied quickly because, to ensure the unfilled materials respond better at this cutting depth, the reinforced materials suffer a slight degradation of quality.

## 1. Introduction

The automotive sector is evolving to a more environmentally friendly methodology. The development in weight reduction and electrification technologies requires substantial changes in material selection and component design [1]. The introduction of polymers and fiber-reinforced materials allows the reduction of emissions and an increase in vehicle material circularity. Thermoplastics are a leading alternative to improve weight reductions and the overall recyclability of vehicles. Carbon fiber reinforced polymers (CFRP) and glass fiber reinforced polymers (GFRP) have become more diffuse in the industry [2], due also to innovative methods of using them, such as rapid prototyping [3] or when machining is a valid alternative to traditional plastic processing for small batch production. For this reason, these materials need to fully abide by the quality standards required in the sector.

The machinability of thermoplastics and composites has been extensively studied to achieve the required geometry, tolerance, and surface finish of automotive components. Analysis of the scientific literature indicates that several aspects have been proven to be vital in the machining of reinforced thermoplastics. Polymers are considered complex materials because of their low thermal diffusivity and thermal dependency on mechanical properties [4,5]. The high values of the thermal expansion coefficient make it challenging to produce parts without shape deviation. The viscoelastic behavior offers a technological challenge because of the interaction between the strain rate and temperature. As a result, the viscous deformation strongly influences the overall quality of the machined surfaces. The process conditions should be accurately selected in the regime with no scaling/tearing or brittle cracking [6]. The machining efficiency is strictly determined by the chemical and thermo-physical properties of the selected polymer and its reinforcement. It is imperative to correlate the machining performances to these materials, evaluating the effect of the process parameters on the surface integrity [7]. The analysis of cutting forces and surface roughness is crucial to appropriately plan and control the machining operation. The cutting force and surface quality should be investigated as the primary response data to achieve optimal results [8]. Thus, machining is performed at low speed and low material removal rates to provide high dimensional and surface quality [9]. Specific polymers present a high impact resistance and suffer deflections, resulting in shape defects in the machined part due to the contact between the cutting tool and the material surface. A reinforced material has a significant impact on tool life due to its abrasive nature [10]. Regarding fibers, debonding between the matrix and fibers, and delamination during material removal, strongly affect the final surface’s texture [11].

The review of the scientific literature available on the material grindability of reinforced plastics reveals several essential factors. These materials, characterized by high cutting heat, high fiber hardness, and heterogeneous structure, are challenging to process, are vulnerable to cracking, and produce large quantities of dust. The latter usually induce adhesive wear on the grinding wheel and decrease the material removal rate and processing efficiency [12]. Moreover, a clean-cut surface without uncut ends is difficult to achieve because the grinding grains cannot sharply cut the fibers [13]. In the grinding process, a cutting fluid is used to minimize the harmful effects caused by a large amount of heat in the cutting zone. Studying the grinding behavior under different lubricant–coolant techniques is decisive to appraise the surface roughness, grinding force, and specific grinding energy [14]. An alternative grinding approach involves using small diameter abrasive cutters or grinding points either with plain or contoured profiles, which can be used similarly to end mills when trimming/routing CFRP [15]. For grinding, the essential values to estimate are the energy partition for mechanical efficiency and the specific energy spent to remove a particular volume of material [16]. Direct measurement of the temperature distribution is complicated due to the minute contact zone for heat transfer. Alternatively, temperature measurement is more accessible, and these measurements are often used to retrieve the heat flux [17]. The influence of the grinding heat on the machined surface is also analyzed based on the grinding temperature at the wheel contact area to evaluate the temperature at which glass transition occurs and thermal alteration can result [18].

The above facts, combined with the described increase in the demand for improving polymer adoption, create a concrete opportunity to research the grindability of reinforced thermoplastics. Grinding of unfilled and glass-fiber reinforced Polyamide 66 (PA66) was carried out, process efficiency was experimentally calculated and monitored, and the main physical variables were determined. In particular, the forces, energy partition, and specific energy of samples produced, starting from conventional methods (extrusion and machining), were experimentally verified under different grinding conditions and with other grinding wheels. The study of the grinding, limited only to polyamide materials, was transferred from the laboratory experience to a practical scale. The novelty of the present work is that the forces, energy partition, and specific energy of polyamide samples produced starting from conventional methods (extrusion and machining) were experimentally verified for the first time under different grinding conditions.

## 2. Experimental Setup

### 2.1. Material Characterization

Extruded TECAMID PA66 provided in plates by Ensinger GmbH (Nunfrigen, Germany) were used as base materials. Due to its lower water absorption, which creates a higher level of dimensional stability combined with a minor variation in the physical properties due to humidity, Polyamide 66 (PA66) is preferred over Polyamide 6 (PA6) for automotive applications. The addition of glass fiber further reduces the water absorption (Table 1).

Fibers absorb a lower humidity level and increase the material’s crystallinity, suppressing the hygroscopic PA tendency. Their addition to the polymeric matrix also enhances the Young’s modulus, and reduces elongation at break and thermal expansion. Fibers are ideal for improving the machinability of the matrix. A more rigid material responds better to cutting, and small chips break more easily from the tool, reducing the amount of heat in the tool–workpiece system. However, if material becomes too fragile, superficial defects and crack initiations start to appear; thus, materials with over 40% of fibers by weight must be carefully machined. Fiber addition reduces the coefficient of thermal expansion, providing another advantage for precision machining. During cutting, temperatures rise due to the friction between the workpiece and tool system, generating residual stresses that influence the final dimensions of the product. The industrial environment did not allow the local humidity of the materials to be controlled. For this reason, all materials were considered to be saturated in all experimental procedures of the present work. The saturation condition was extended to the other processing processes carried out on all specimens.

Differential Scanning Calorimetry (DSC) was performed on small samples (5 mg) to verify material properties changes because of the storage conditions. A DSC 403 F3 Pegasus (Netzsch-Gerätebau GmbH, Selb, Germany), equipped with a silver furnace, performed a thermal cycle from −50 to 340 °C at a rate of 20 °C/min. The glass transition occurred around 45–50 °C, and the wide variation to the rubbery state was completed inside the expected range (Figure 1). The glass transition temperature of both materials suffered no significant variations due to stocking.

### 2.2. Grinding Machine and Sensors

The primary grinding parameters were the input CNC variables as the starting point of any process analysis. Their influences were evaluated with the coolant and the grinding wheel conditioning to estimate all physical effects on the sample and the machined piece’s final quality. The selected kinematic parameters were not able to provide high quality if the cooling system was incorrect or the grinding wheel’s dressing was not optimal. These errors led to excessive temperatures on the surface, causing burns and high residual stresses. Defects were not the only phenomena tied to the monitoring of physical variables during machining. Considering that the volume of material removed by grinding was smaller than that of other manufacturing processes, a significant amount of energy was spent to achieve a particular tolerance for a product. Observing the mechanisms while grinding a product enabled the evaluation of the process’s efficiency. Effectively describing the quantity of energy spent to grind a piece, and the source of energy losses, may result in changes in the entire production line.

The signal acquisition framework consisted of several modules, as Figure 2 shows. The data acquisition system (DAQ) on a Planomat 408 machine (Blohm Jung GmbH, Hamburg, Germany) with a SINUMERIK 840d sl CNC system allowed sampling the power signal directly on the grinding wheel spindle. The maximum capacity reached 1450 rpm and the maximum spindle power was 7.5 kW. The DAQ had a sampling rate of 0.002 s with no embedded filters. In addition to the command system, the holder’s design allowed the thermocouple’s insertion into the sample while permitting the sample to directly contact the force measuring surface. The holder blocked all movements and provided a preload for the force sensors. Two loading piezoelectric cells (Kistler Group AG, Winterthur, Switzerland) acquired the normal and tangential forces during grinding (see Table 2 for the main characteristics). This cell was chosen due to its fast response to force variations, suitability for machining purposes, and calibration for the application.

Because the magnitude of the polymeric grinding forces was not initially known, adapting the preload was essential. A pressure plate sensitive to the normal and tangential forces was realized with a regulating bolt to control the sensors’ preload. The bolt was tightened with 20 N×m of torque, and the force sensors detected weights from 0.5 to 5 kg with decimal precision to measure the forces during cutting. A MAXYMOS TYP5877A/B (Kistler Group AG, Winterthur, Switzerland) acquired the cells’ signals, working as an amplifier and monitoring system, with sampling of 0.001 s. The applied filter was 5 Hz. The bolt and pressure plate accommodated a Type J thermocouple (1.0 mm diameter and 500 mm length) to measure up to 760 °C. The thermocouple measured the workpiece temperature, and a Multicon CMC-99 controller (Simex Sp. z o.o, Gdansk, Poland) recorded the logged data with a sampling interval of 0.1 s. The temperature sensor’s tip was 29.5 mm from the pressure plate top, and the hole was filled with epoxy to fix it. The temperature had to be filtered as the power after the data acquisition in the post-processing.

### 2.3. Grinding Parameters and Surface Evaluation

A stationary diamond plate, having dimensions of 0.6 × 0.6 × 5 mm^3^, equipped with 3 tips, was used to dress the grinding wheel with a low-pressure cooling with an emulsion CIMCOOL CIMTECH A31F oil plus water at 3.5% in volume. The grinding machine had two cooling nozzles. The first nozzle, oriented to the contact point between the wheel and the workpiece, was responsible for controlling temperatures during machining. The second nozzle, oriented to the rear of the wheel, cleaned the grinding wheel of any debris accumulated in the porous structure, that may have caused non-cutting situations and temperature increases. In addition, low pressure (3 bar) and high pressure (6 bar) cooling settings were available.

As demonstrated in previous research, aluminum oxide (Al_2_O_3_) grain wheels were selected due to their availability, price, and compatibility [19]. Two wheels, coded 45A120-5G11RM-LV233/35 and 87A80-2H13RM-JV56/35 (ELBE Schleifmittelwerk GmBh & Co KG, Sachsenheim, Germany), had an initial diameter of 406 mm and a thickness of 30 mm.

The main difference between the two grinding wheels was their grain size, equal to FEPA120 (finer) and FEPA80 (fine) grit, respectively, for the first and second wheel. As specified in the code, the second wheel had a more open, porous structure, resulting in a harder overall tool, and the first wheel had a more closed structure with less porosity, and was softer. Figure 3 shows the entire setup. The CNC code set the pressure plate as the relative zero before data acquisition. A leveling pass for all samples guaranteed the exactness of the cutting depth to the experiment’s value. The process always started with wheel dressing to remove any contaminants from the previous test. The cutting tool was then aligned with the sample, and machining started. The coolant application depended on the type of experiment being performed, and was completely cut off on demand. After the wheel’s positioning, the previously described temperature and force sensors were activated, and their respective acquisition systems gathered data. At the end of each step, the surface quality was evaluated, and the samples’ microscopic alterations were observed. The pieces were analyzed immediately after processing with an Evo MA25 Scanning Electronic Microscope (Carl Zeiss AG, Oberkochen, Germany). The surface roughness was then measured using a MarSurf XCR20 surface contour and roughness unit (Mahr GmbH, Göttingen, Germany). This instrument, with a resolution of 0.19 µm using the 175 mm MFW250 mechanical probe arm, and 0.04 µm relative to the measuring system, operated at a speed of 0.5 mm/s and a measuring distance *L_s_* of 3.2 mm in the normal direction to the grinding path.

## 3. Results

Preliminary analyses compared ground and nonground specimens. From rough materials, 60 samples for each material (120 in total) with dimensions of 60 × 30 × 5 mm^3^, were realized and their surface quality was characterized before grinding. *R_a_* and *R_z_* were measured (Table 3), testing the surface roughness for normality with the Anderson–Darling technique. Quality control was enforced to decide to accept a part when the print requirements were not consistent with measurements on the surface gages in the local facility. In these cases, *R_a_* and *R_z_* were the fastest parameters to be determined. More in-depth analysis should be made offline with the use of focus-variation microscopy [20]. Compared to the glass-fiber PA66, the unfilled PA66 presented a lower mean value but a higher deviation, and failed to fit a normal distribution, as observed in the *p*-value. The deviation was lower for the PA66GF30, and there was no significant departure from normality because the *p*-value for *R_a_* and *R_z_* was higher than 0.05. The presence of fibers and material elongation at break accounted for this behavior. The presence of fibers generated a rougher surface due to the randomness of their distribution in the matrix. During machining, there was also a certain degree of adhesion between the removed layer of material, tools, and the surface, leading to defects such as fiber pull-out and fiber/matrix debonding, negatively impacting the surface texture. For this reason, the surface quality of PA66GF30 was worse than that of PA66. Alternatively, the lower deviation was explained through the significant difference in the deformation that the materials presented when breaking.

The PA66 deformed at 70% of its original length, compared to 14% for the filled PA66. This property generated more minor chips during manufacturing, less heat accumulation on the cutting zone, and a more reliable process. Less temperature in the contact zone led to more desirable material behavior because the material was glassy and presented a lower degree of viscous deformation below *T_g_*. Some preliminary experiments suggested three initial cutting depths *a_e_*, repeated three times. A finishing grinding wheel completed the tests to achieve high quality with one grinding pass. As a result, the total number of samples per material was nine. Table 4 reports the dressing and grinding parameters. At the beginning of each test, the grinding wheel was dressed before the data acquisition pass. During dressing, the cutting speed was higher than the normal operating speed to remove all traces of previous experiments. Excessive wear of the grinding wheel resulted if the dressing speed used was tool low. Each leveled sample, starting with the same flatness, avoiding influences from other processes. Due to the different data acquisition channels, the original data were not synchronized, and a post-processing phase fused all the data. The selected grinding parameters for this experiment diverged from those of previous works [19,21] due to the sample geometry and sampling limitations. The length of the grinding wheel path for force acquisition was only 60 mm. High infeed speeds would provide insufficient data for the acquisition. This also happens if the workpiece speed is too low. The number of sampling points would be too high for the instrument, and forces may become so low that they would not be sensed by the loading cells.

Several phases were identified during acquisition (Figure 4):
There was no contact between the sample and the tool. There was no contact with the workpiece, and the spindle wheel used minimal power to maintain the imposed cutting speed *v_c_*.The coolant was activated, generating a significant variation in the tangential force and a minor variation in the normal force. The tangential component was more critical because the coolant nozzle pointed directly at a point close to the interaction between the workpiece and the wheel. Up-grinding was performed, and the wheel spin facilitated the entrance of the coolant into the contact zone. The spindle wheel continued to maintain the required cutting speed *v_c_*.The first contact between the tool and specimen caused an increase in both forces. The drive increased power to the spindle to maintain *v_c_*. The grinding area was at the maximum level, and the power entered a steady state at a higher power level. The difference between the idle power and this level was the net grinding power responsible for the cut, and was directly linked to the tangential force.Contact remained stable. The normal force maintained the pressure in the workpiece while the tangential force cut the material. The tangential force decayed with the material removal.A rapid reduction of both signals appeared. The wheel began to move away from the piece and, because there was no more impediment to spinning, the power decreased.The process is finished. The normal signal was not able to return to zero because of a spring back compensation due to the compression forces during machining. Grinding was finished, and the power returned to idle.

Observation of the normal and tangential forces (Figure 5 and Figure 6) shows that both forces grew with increased cutting depth. The reason for this behavior was an increase in the volume of material removed with an increase in the cutting depth. The DAQ tool monitored the cutting phenomena and the sample–tool interactions during machining. Some basic phenomena were detected. One was the exact position of the thermocouple hole in the acquisition of all data, detected by the slight deflection of the normal force values along the plateau. The second was that the wheel lost sharpness due to excessive material deformation, represented by a sudden force increase. The behavior of PA66GF30 was like that of PA66, showing an increase in force with an increase in the cutting depth. The normal forces were steadier and showed a smaller decrease in process efficiency. The tangential forces did not appear to be significantly affected by the presence of fibers. Another favorable aspect to examine was the spring back compensation of the normal force. The filled PA66 was less sensitive to an increase in cutting depth than the PA66. No significant differences existed for the two grinding wheels, with the exception of curves with *a_e_* equal to 150. The decay in the tangential forces in all cases can be associated with the influence of the cutting fluid in the process. Because the nozzle is pointed in the same direction as the wheel moves, the amount of fluid directly applied to the surface reduces. However, the points where the wheel engages and disengages the piece are seen in the literature [22,23].

The main difference between power and force acquisition was the idle power, defined as the power spent during the spindle operation in a steady state before machining. There was no response to the application of the coolant because it happened in the tangential force. The grinding power *P* is associated with the tangential force component *F_t_*, given by [16]:(1)$$P=v_{c}\times F_{t}$$

This value was compared with the total energy given by the machine’s spindle while cutting *P_total_*, and provided the energy partition *ε_p_* and the lost power *η* [20]:(2)$$\varepsilon_{p}=\frac{P}{P_{total}}$$
(3)$$\eta =1-\varepsilon_{p}$$

Results of the energy partition versus cutting depth are presented in Figure 7. The grinding power increased with an increase in the cutting depth. Compared to the tangential forces, the cutting power values provided a better insight into the process behavior.

All gathered data of each sample were averaged to represent each cutting depth data point. Only for one set (PA66GF30, 45A wheel, and 150 µm), an increase in cutting depth denoted a loss of cutting partition and lost power increased. Energy partition was expected at some point as the cutting depth increased, because its increase caused the grains to be wholly immersed in the material that was to be removed, thus not providing an opportunity for the cutting edges to properly machine the surface. Moreover, the non-cutting surfaces, e.g., the bonding material, came into contact with the workpiece and increased the amount of rubbing and heat in the contact zone. In addition, there was no loss of power in the latter case, only a loss of energy partition because the process for these specific conditions reached a technological limit. The power at the spindle continuously increased because the machine complied with the parameters set by the operations. However, the value to effectively perform grinding changed. At that stage, the grains were too small for the cutting depth. They were not converting the spindle power as was possible at the other cutting depths and, instead, most of the energy given to the system was wasted as heat. This heat was directed to the grinding wheel–fluid system, rather than the workpiece, as evident from the temperature measurements and plowing of particles present in the cutting zone.

The decrease in lost power suggested that the process forces were dominant. As explained later, the power may be lost in other operations, such as plowing, chip formation, and temperature generation on the surface. Combining a high Peclet number (range 116–201) with optimal cooling did not allow for a large quantity of heat to be absorbed and passed to the workpiece. The Peclet number *P_e_* is given by:(4)$$P_{e}=\frac{v_{w}\times l_{g}}{4\times \alpha }$$
where *v_w_* is the infeed speed; *l_g_* the geometrical contact length between the workpiece and the wheel, given by *l_g_* = *a_e_* × *D*; and α the thermal diffusivity [24]. Most grinding operations are undertaken with *P_e_* equal to 1, balancing the convective transport with heat transport [22]. A reduction of the thermal diffusivity *α*, typical of a polymer, caused an increase in *P_e_*. The physical meaning was a reduction of the amount of heat transferred to the workpiece. No sensitive variations of temperature affecting the surface integrity were recorded; this was also the case for the highest cutting pass. The temperature in the experiment, with the cooling applied, did not present a significant increase, with a maximum variation of 5 °C. The direct contact between the grains and the workpiece generated heat. However, the advective transport rate was more effective than the diffusive transport rate. Most of the thermal energy entering the workpiece was transmitted to the coolant–grinding wheel system. The specific energy is:(5)$$e_{s}=\frac{F_{t}\times v_{c}}{b\times \alpha_{e}\times v_{w}}$$

Results of the specific grinding energy *e_s_* are presented in Figure 8. As previously discussed, the tangential force *F_t_* increased with the cutting depth. However, the ratio *F_t_*/*a_e_* decreased because the increase in the cutting depth was more significant, with the other parameters in the equation being constant. Thus, the specific energy *e_s_* fell with a rise in the cutting depth. The overall trend was a reduction in the amount of energy to remove the material with increased cutting depth. The machining of PA66GF30 required less energy than that for PA66. The brittle nature of the composite promoted this behavior. An increase of 50 to 100 µm had a greater impact than an increase of 100 to 150 µm. The cutting depth increased three times, but the specific energy did not decay in the same order. This was another indicator that, for polyamides ground in these conditions, the specific energy converged to a value around 50 to 60 J/mm^3^. As the figure shows, an increase from 50 to 100 µm caused a reduction in specific energy by 50% for all cases, whereas a rise from 100 to 150 µm resulted in a 30% reduction.

The surface quality indicators *R_a_* and *R_z_* enlarged the result analysis (Figure 9). The two roughness parameters acted in an opposing manner with the variation of *a_e_*. *R_a_* is a measurement of surface integrity because it is an arithmetic average of the entire sampling length. This measure provided a general idea of the quality of the surface, but averaged out most of the defects present in the specimen. In contrast, *R_z_* is an average of the five lowest valleys and the give highest peaks. This measure increased with an increase in the number of ploughing and rubbing zones. In this work, observation of both parameters was necessary because surface integrity is essential for materials this soft, and the fast evaluation of defects is essential to real-life production. Evaluating the surface with only one of these parameters would result in a disagreement between the results found from SEM, which showed a more damaged surface, and the resulting surface roughness. An increase in cutting depth to 150 μm produced a reduction in *R_a_* but an increase in *R_z_*. The grinding forces on ground specimens increased with an increased cutting depth, causing a decrease in surface quality caused by higher wheel wear. The workpiece speed increased, resulting in an increase in the plastic deformation and surface roughness. The larger the grinding depth and the longitudinal feed amount, the larger the plastic deformation, leading to a higher the surface roughness value. The grinding wheel has a high temperature, and the heat was dominant, so the cutting fluid was crucial. The use of cutting fluid reduced the temperature of the grinding zone and burns, and washed away sand and chips, thus preventing scratching of the workpiece and reducing the surface roughness value. The surface quality surpassed the results obtained by milling with a single pass of grinding. The surface quality of unfilled PA66 diminished with this procedure, and, curiously, it presented the same reduction at the highest level of the cutting depth as the previous experiment. Even at a controlled temperature, the material viscosity can interfere with its quality during machining operations. On the contrary, the filled PA66 better responded to grinding because of the presence of fiber. The volume of removed material was constant with the increase in the cutting depth because more grains reached the contact zone. Consequently, the groove depth on the ground surface left by each grain was smaller, improving the surface roughness. This was caused by the material’s resistance to its removal due to the high removal rate during one pass.

The SEM image of the surfaces enables a more in-depth analysis. As Figure 10 shows, the amount of plastic flow and the incomplete cut for PA66 was more evident with an increase in cutting depth. Some rubbing zones existed where grains not aligned with the cutting height had little interaction with the surface. As the depth and the number of grains engaged in the zone increased, the material was not perfectly cut, leaving burrs and ploughing zones [25].

The grains did not engage with the workpiece at the same height, despite dressing and truing the grinding wheel. Moreover, they had stochastic positions, angles, and dimensions—the specified granularity guaranteed only a specific range of dimensions. The grains had a certain tendency to break themselves during machining and form new cutting points.

For the filled PA66, the same type of surface defects was observed (Figure 11). The fibers were almost all fractured and, as the cutting depth increased, they appeared to slowly position themselves along the cutting direction. The lower deformation at break possessed by the material facilitated a smoother ground surface, mainly in the matrix zone, and resulted in better surface quality of the samples. After grinding, the short fibers only presented fractures and a reduced number of pull-outs. Because the chosen parameters values were close to the fiber dimensions’ lengths, the process did not affect the surface roughness’s overall results. Observing all the SEM images, the lower cutting depths produced smoother zones (cutting), and the higher cutting depths showed a greater propensity to defects. The cutting wheel started with a very high volume of material to remove, the strain applied to the samples was higher, and the plastic deformation produced more burrs and ploughing.

The efficiency of the process was attributed to the low amount of heat lost to the piece due to the low diffusivity and the application of the coolant. Because the material temperatures remained stable below the glass transition, the resulting mechanical properties were more suitable for grinding. A less ductile material was removed more efficiently and produced more minor defects from plastic deformation. Another indication was the better surface quality of the filled materials. Although energy partition increased for all cases, with the exception of one, this behavior was expected to change while also observing the specific energy of the process. As *e_s_* converges to a value, it can be interpreted as a constant for the material. In addition, because the forces continuously increased as the cutting depth increased, the value of energy to cut one cubic millimeter of material remained the same, and the forces were used for different physical mechanisms.

## 4. Conclusions

This work studied the grinding of polyamide samples by identifying the essential physical mechanisms involved in the process. Energetic and efficiency evaluation was performed by comparing the process events with the final quality of the ground surface. The temperature, forces, and power monitoring led to non-conventional cutting depths above 100 µm and an increased material removal rate, while also yielding good surface quality results. The cutting forces’ behavior was monitored as a function of the cutting depth. The results were combined with the spindle drive’s power acquisition to assess the efficiency of cutting polyamide. In contrast to metallic materials, polymers presented a different distribution of energy within the grinding process. Most of the energy was converted to the strain and cutting of the piece, resulting in an efficient operation even with cutting depths that were three times greater than the production parameters for conventional materials (metals and ceramics). The specific grinding energy stabilized around 60 J/mm³ for both materials at high cutting depths, indicating that a further increase would probably affect the energy partition. Strains were applied quickly, and the reinforced materials slightly degraded the quality to ensure the unfilled material responded better at this cutting depth. Large deformation prior to rupture showed an increase in the roughness. Making the material as brittle as possible, via surface temperature control and the addition of fiber, was found to produce favorable results. The SEM images demonstrated that an increase in cutting depth provided more grain engagement with the piece, but not always a good surface quality. In all cases, the fibers presented good bonding with the matrix before and after grinding. When the fibers were ground, they appeared to tend to crack rather than pulling out or debonding. Thus, grinding by utilizing high cutting depths (over 100 µm) is a serious contender as a finishing process for most production lines because it can efficiently and rapidly remove a large volume of material.

Future research will extend the monitoring and evaluation of the cutting speed variation to assess its influence on surface quality and process efficiency.

## Figures and Tables

**Figure 1 materials-14-05041-f001:**
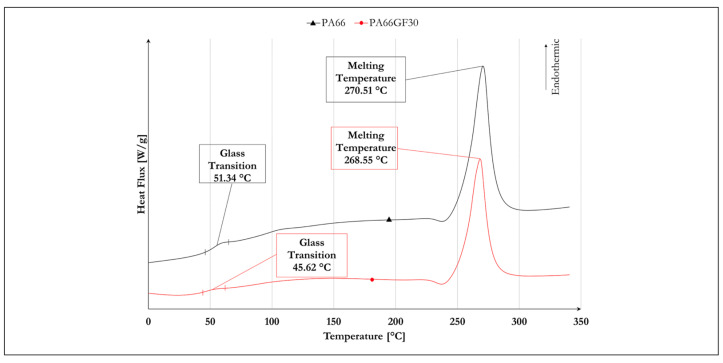
DSC of PA66 and PA66GF30.

**Figure 2 materials-14-05041-f002:**
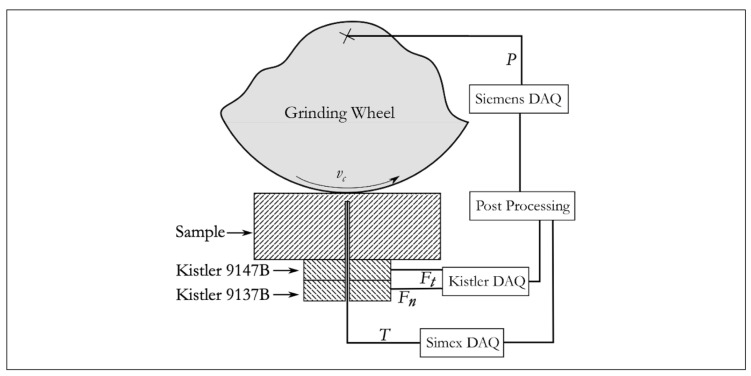
Diagram of the sensors and data acquisition systems.

**Figure 3 materials-14-05041-f003:**
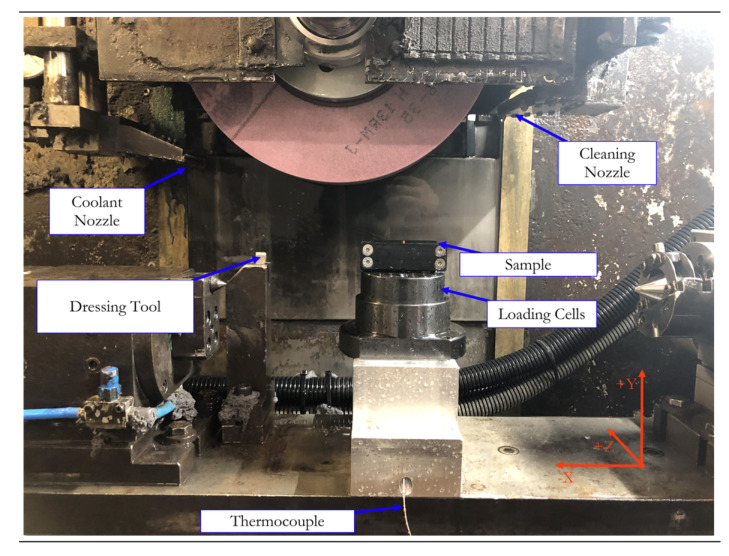
Internal details of the grinding setup.

**Figure 4 materials-14-05041-f004:**
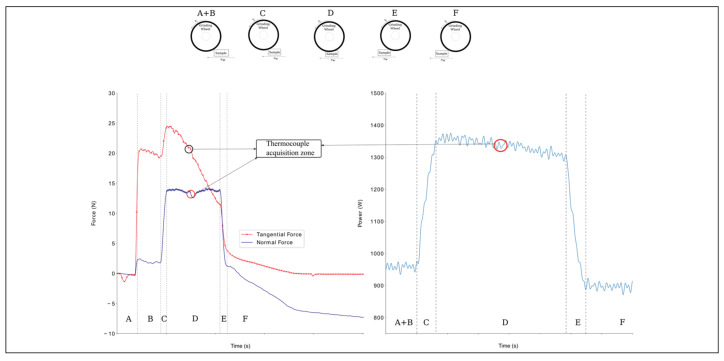
Examples of data acquisition during the grinding process.

**Figure 5 materials-14-05041-f005:**
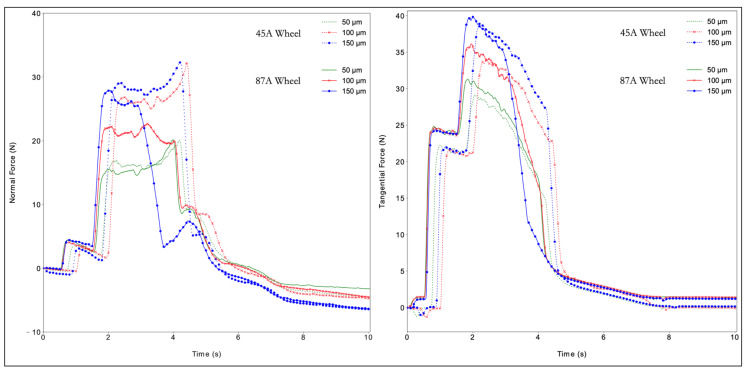
Normal (**left**) and tangential forces (**right**) for PA66.

**Figure 6 materials-14-05041-f006:**
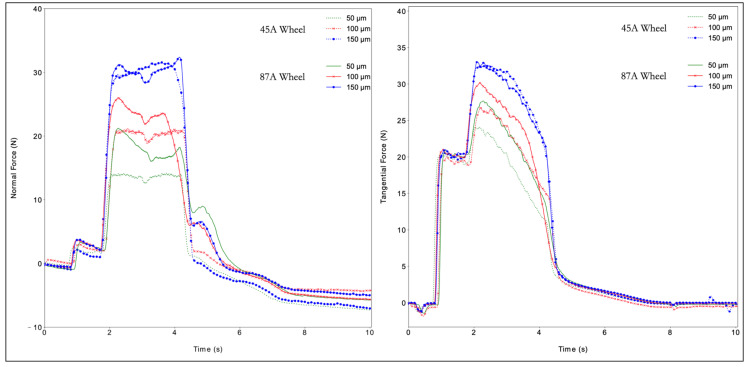
Normal (**left**) and tangential forces (**right**) for PA66GF30.

**Figure 7 materials-14-05041-f007:**
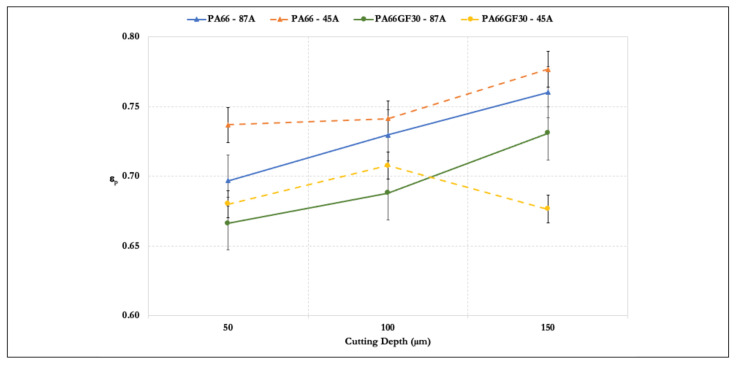
Energy partition vs. cutting depth.

**Figure 8 materials-14-05041-f008:**
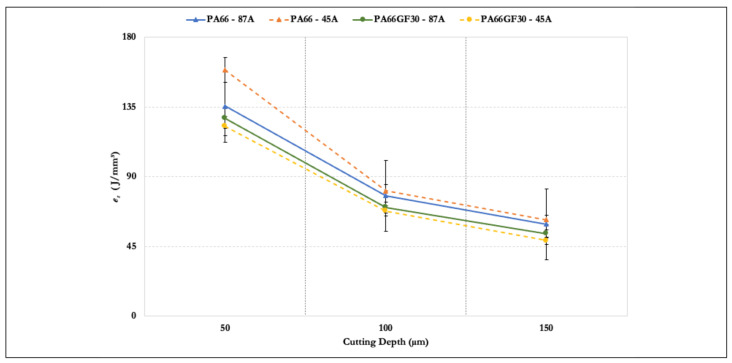
Specific grinding energy vs. cutting depth.

**Figure 9 materials-14-05041-f009:**
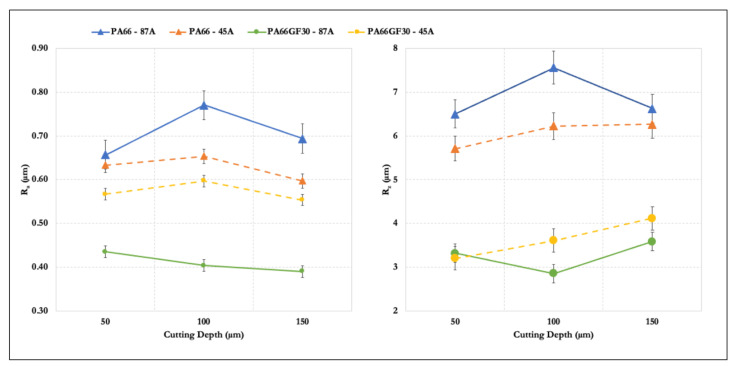
*R_a_* (left) and *R_z_* (right).

**Figure 10 materials-14-05041-f010:**
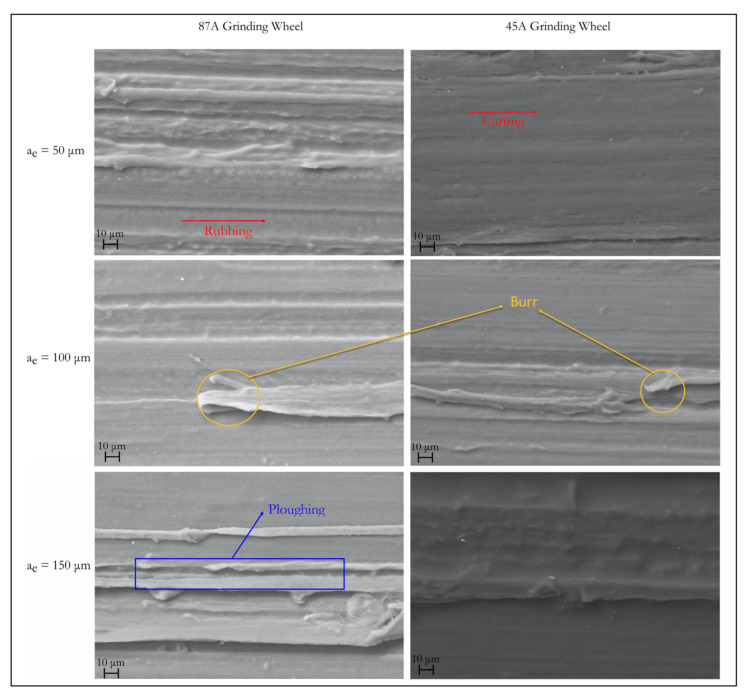
SEM images of the PA66 ground surfaces (500×).

**Figure 11 materials-14-05041-f011:**
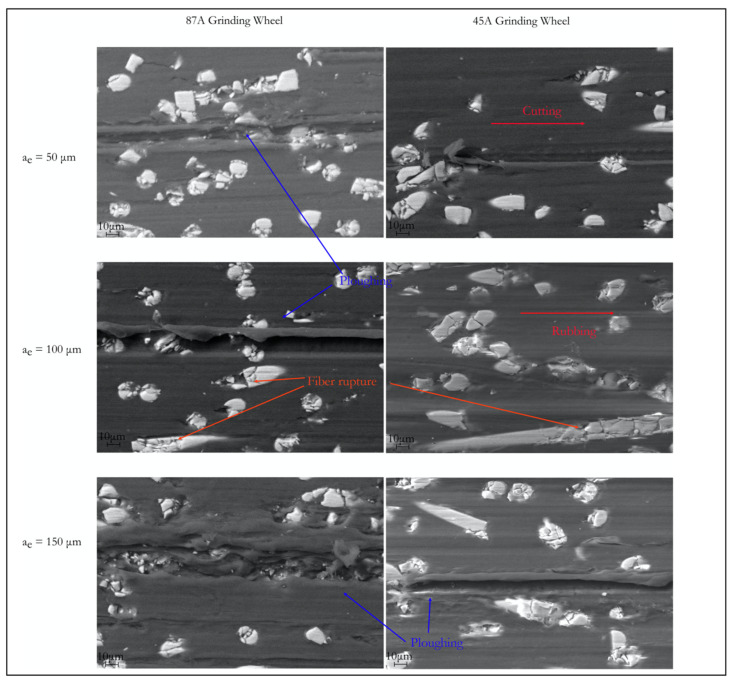
SEM images of PA66GF30 ground surfaces (500×).

**Table 1 materials-14-05041-t001:** Main properties of the investigated PA66 materials (from supplier datasheets).

	PA66	PA66GF30	
Property	Value		Unit
**General**			
Material class	PA66	PA66	-
Reinforcement (glass fiber)	0	30	%
**Physical**			
Density (*ρ*)	1.15	1.34	g/cm^3^
Glass transition temperature (*T_g_*)	47	48	°C
Melting (Softening) temperature (*T_m_*)	258	254	°C
Thermal conductivity (*k*)	0.36	0.39	W/(K × m)
Thermal diffusivity (*α*)	0.20	0.24	mm^2^/s
Water absorption (24 h/96 h–23 °C)	0.2/0.4	0.1/0.2	%
**Mechanical** (**Tensile test**)			
Young’s modulus	3500	5500	MPa
Yield tensile strength	84	91	MPa
Ultimate tensile strength	85	91	MPa
Elongation at break	9.6	2.1	%
Coefficient of thermal expansion	120	50	µm/m/°C

**Table 2 materials-14-05041-t002:** Loading cell properties.

Sensor	Component	Range (kN)	Sensitivity (pc/N)
9137B	Normal	0–80	3.8
9147B	Tangential	0–8	0.1

**Table 3 materials-14-05041-t003:** Statistical values of the roughness of the milled samples (N = 60).

	PA66	PA66GF30	
Property	Value		Unit
***R_a_***			
Mean	0.42	0.72	µm
Std. Deviation	0.16	0.08	µm
*p*-value	<0.005	0.367	µm
***R_z_***			
Mean	2.04	4.01	µm
Std. Deviation	0.62	0.44	µm
*p*-value	<0.005	0.152	µm

**Table 4 materials-14-05041-t004:** Parameters of dressing and grinding.

	Dressing	Grinding	
Property	Value		Unit
Cutting Speed (*v_c_*)	4000	3000	mm/s
Z-axis speed (*v_z_*)	120/400	0	mm/min
Infeed speed (*v_w_*)	-	1500	mm/min
Cutting depth (*a_e_*)	36	50/100/150	µm
Number of passes	1	1	-

## Data Availability

Data avalible on request. No public dataset.

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
