# Peer review of "Forces and Specific Energy of Polyamide Grinding"

_materials, 2021, doi:10.3390/ma14175041_

Round 1

Reviewer 1 Report

On the axis of tangential force and grinding power - Figure 4. Examples of power, tangential force and normal force data acquisition during the grinding process - no value, and the dependencies are the result of measurements, similar to Figures 5 and 6. Similarly in Figure 1. DSC of PA66 and PA66GF30, enter the values on the temperature axis.

In Table 3. Dressing Parameters, Parameter Value Unit Grinding Wheel Speed 40 m / s - and in Table 5. Deep grinding parameters. Process Parameter Value Unit Cutting Speed (vc) 30 m / s. Whether there was a change in the speed of the grinding wheel for dressing.

It is worth answering the question why the specific energy for grinding with a 45A grinding wheel (grain size 120 micrometers) is lower than the energy for 87A (grain size 80 micrometers).

Many parameters were probably determined in the measurements of the geometrical structure of the surface. Limiting oneself to Ra and Rz causes the omission of important information, for example, on features of elevations and horizontal features. 

Author Response

See comments in the pdf file

Reviewer 2 Report

1) In introduction section, the literature review is too general and some relevant papers dealing with the investigation on grinding of polyamide are missing. Therefore, authors should revise it and state clearly why their work is relevant, and which the expected advances in knowledge are.

2) It seems that the novelty of the conducted research is not very strong.

3) In the present article, it is necessary to clearly indicate the specific references and justifications for choosing the deep grinding parameters (cutting speed, infeed speed, cutting depth, grinding thickness, etc.)

4) The author wrote that “By observing the results of the normal and tangential forces (Figs. 5 and 6), both forces grow with an increase of cutting depth.” It is necessary to provide the reasonable and adequate explanations for the above experimental results.

5) “The grinding power increased with a rise in the cutting depth.”

“Only for one set (PA66GF30, 45A wheel and 150 µm), an increase of cutting depth denoted a loss of cutting partition and lost power increased.”

Similarly, it is necessary to provide the reasonable and adequate explanations for the above research findings.

6) In this article, it is necessary to reasonably and adequately explain why the specific grinding energy decreases with the increase of cutting depth.

7) In Fig. 9, it is necessary to comprehensively compare and analyze the effects of cutting depth on Ra and Rz, and provide the reasonable and sufficient explanation for the experimental results.

8) “An increase of cutting depth to 150 µm produced a Ra reduction but a Rz increase.” Similarly, it is necessary to provide the reasonable and adequate explanations for the above research findings.

9) In Fig. 10 and Fig. 11, it is necessary to comprehensively compare and analyze the influences of cutting depth on the ground surface morphology of PA66 and PA66GF30, and provide the reasonable and sufficient explanations for the experimental results.

10) The abstract and conclusion section need to be improved.

11) It is found that the manuscript is incomplete in all sense and the discussion section is missing.

12) The results are mainly presented by figures and tables. It is more important to provide the sufficient and reasonable explanation for the research results. This is also the weakest aspect of the study.

13) The author needs to discuss all the research results in more detail and give more reasonable and sufficient explanation for all the research findings.

14) The limitations of the study are not considered.

15) The format of all the references should be modified according to the journal guidelines.

Author Response

Please see comments in the pdf file.

Reviewer 3 Report

Abstract: The abstract is expected to include a brief digest of the research, that is, new methods, results, concepts, and conclusions only. The abstract needs to be more focused and achievements needs mentioned clearly. At the moment abstract is more like an introduction than abstract. Please add some information from the conclusion (quantifications).

Introduction based on old references. I personally feel that this part of paper is not concise enough from a reader’s perspective. Introduction must provide a comprehensive critical review of recent developments in a specific area or theme that is within the scope of the journal (advanced manufacturing), not only a list of published studies or a bibliometric one. Introduction is expected to have an extensive literature review followed by an in-depth and critical analysis of the state of the art. References section should be extensive about information connecting with surface quality. I suggest add information to better describe what other researchers have done in this area. Please look on papers published by prof Kaplonek or prof Nadolny. I suggest add important and new articles from this field, for example:

The Use of Focus-Variation Microscopy for the Assessment of Active Surfaces of a New Generation of Coated Abrasive Tools By: Kaplonek, Wojciech; et al.  MEASUREMENT SCIENCE REVIEW  Volume: 16   Issue: 2   Pages: 42-53   Published: APR 2016

The strengths and limitations of the applied approach should be clearly identified for the readers of the paper.

Why Authors used PA66?

All variables parameters should be written in italic style.

The discussion is shallow and needs more details, the observations and future trends. This chapter should be connected with others published papers.

Some of the bullet points on the conclusion are simplistic;  Please try to emphasize your novelty, put some quantifications, and comment on the limitations. This is a very common way to write conclusions for a learned academic journal. The conclusions should highlight the novelty and advance in understanding presented in the work.

Author Response

Please see comments in the pdf file.

Reviewer 4 Report

  • Abstract: Please comment on the results of cutting energy. Also, comment on the difference in cutting between the filled and unfilled materials
  • Thorough English revision is required. Please make sure you revise your manuscript by a native speaker or professional editing service
  • The novelty of the work and research gap must be identified clearly at the end of the introduction
  • Between line 67-80, you have made lot of statements, some of them need references. This paragraph also does not look like description of methods. Please move this to introduction or discussions.
  • Table 1: if you have taken these data from supplier’s data sheet you need to make this clear.
  • Table 3: Do you mean dressing feed/depth. Z Axis speed. What is this? You have not shown axis system of the grinding set-up
  • Line 149: Mention the sensors used to collect what data.
  • Table 5: Do you mean grinding wheel thickness?
  • Figure 4 caption: give short description of A, B, C etc
  • Figure 5 and 6 captions: avoid using left right. Instead use (a) and (b)
  • In section 2 you should add a subsection describing the statistical analysis
  • Figure 7 and 8: vertical axis full name of the parameters needs to be included with their units.
  • Some organization changes are required. In the results and discussions you do not put the experimental parameters. They should be in the experimental section. For example table 5. Therefore, anything relevant to procedure you need to move to section 2
  • Conclusions: This needs to be re-written with specific results on the effect of different grinding wheels, different materials on surface finish, wear behaviour, energy partition, cutting energy etc.
  • Discussions also must be improved again considering grinding wheels, materials, cutting depth

Author Response

Please see comments in the pdf file.

Round 2

Reviewer 2 Report

The authors have considered some comments of reviewers. Nevertheless, the explanations for the following questions are not reasonable and sufficient. To improve the quality of the paper, the following issues need to be addressed.

Q5. “The grinding power increased with a rise in the cutting depth.”.“Only for one set (PA66GF30, 45A wheel and 150 µm), an increase of cutting depth denoted a loss of cutting partition and lost power increased.”

It is necessary to provide the reasonable and adequate explanations for the above research findings.

Q6. In this article, it is necessary to reasonably and adequately explain why the specific grinding energy decreases with the increase of cutting depth.

Q7. In Fig. 9, it is necessary to comprehensively compare and analyze the effects of cutting depth on Ra and Rz, and provide the reasonable and sufficient explanation for the experimental results.

Q8. “An increase of cutting depth to 150 µm produced a Ra reduction but a Rz increase.” Similarly, it is necessary to provide the reasonable and adequate explanations for the above research findings.

Generally, the wear of the grinding wheel leads to an increase in surface roughness. But, the author wrote that “The grinding forces on ground specimens increased with an increased cutting depth, causing a surface roughness decrease caused by higher wheel wear.”

Therefore, it is necessary to provide the more reasonable and adequate explanations for the above statement.

Author Response

See attached pdf file.

Reviewer 3 Report

Congratulations, the paper is ready for publication

Author Response

Thanks for your precious support.

Reviewer 4 Report

No further changes are required.

Author Response

Thanks for your precious support.

Round 3

Reviewer 2 Report

In Fig. 8, the author wrote that “The overall trend was reducing the amount of energy to remove the material with increased cutting depth.”

But, the author did not provide the reasonable and adequate explains for the above research findings.

Author Response

The reply to the open question is attached to the pdf file.
